

# MORAL LINES OF CREDIT:

# FORGING RACE PROJECTS, CITIZENSHIP, AND NATION ON ONLINE U.S.

# SPOUSAL REUNIFICATION FORUMS

G.M. Longo

**Virginia Commonwealth University**
Longog2@vcu.edu

Monday, June 27, 22

## Abstract

This study investigates how U.S. citizens petitioning for "green cards" on behalf of foreign national spouses uphold the U.S. racial project as they navigate the spousal reunification process. It also explores the role of online communities as crucial "brokers" and mediators between citizen, noncitizen, and the state. This work troubles the dichotomy between immigration officers/couples while giving primacy to the citizen-spouse's voices. Using content analysis of an online forum where petitioners exchange advice with similar others, I show the citizen's complicity with the racialized hierarchical order of the American nation. Ultimately, family migration policies and regulations are exercises in state-building, and nation-building, and citizens partake in it while trying to secure their own family, disciplining themselves to align with the state's ideal of what a proper future nation should look like.

# Introduction

"I thought because I am an American citizen, the government would conveniently deliver [my husband] to me with a big red bow...Nope. You'll have to earn it."- Goldberry1313, Immigration Pathway[i], forum member.

Any U.S. citizen can petition for an immigrant visa on behalf of a foreign partner, allowing them to cross the border, secure employment, and naturalize quickly. However, couples must submit evidence to officials demonstrating that their relationships are "valid and subsisting" (USCIS 2013) and satisfactorily address any aspects of the relationship that indicate marriage fraud such as large age difference, sending money, and cultural differences (i.e., "red flags") to gain approval. Officials do not disclose reasons for denials or what constitutes convincing proof for U.S. national security purposes (Rogers 2012). Nevertheless, U.S. petitioners must decipher the meanings of these requirements anyway.

This study investigates how U.S. petitioners engaged in the spousal reunification process perpetuate the U.S. racial project via intersectional family norms. The state is interested in upholding its racial projects and controlling its territories, resources, and authority (Bonizzoni 2019; de Hart 2015; Macklin 2022), so the literature tacitly fosters a "bad" immigration officer/ "good" migrant couple dichotomy as scholars focus upon the role of state actors in the evaluation process. However, it is essential to look at the petitioners' engagement and the parts that the process forces them to play to understand better how racial projects perpetuate at a systemic level.

I introduce the concept of a moral line of credit to contextualize petitioners' roles and strategies to negotiate the state's moral scrutiny. Citizens draw on their moral line of credit by mobilizing combinations of 'character status assets,' such as race, gender, and class identities, and performances of hegemonic family norms to showcase their worthiness to immigration officials to "earn" the privilege of visa approval.

I turn to the conversation archives of Immigration Pathways (IP). In this free self-help immigration forum, U.S. petitioners clarify U.S. criteria and exchange advice with others with successful petitions. Because women's partners are often more scrutinized than men's, I analyze conversation threads from two regional sub-forums on IP primarily populated with white and Black U.S. citizen women: The Sub-Saharan Africa forum (SSA), where members are petitioning for regional Black men, and the Middle East/North Africa (MENA) forum, where participants petition for Arab men.

I show that women petitioners interpret U.S. immigration policy requirements and "red flags" with intersectionally organized racial ideologies, behaviors, and expectations about the U.S. hegemonic family. I argue that the U.S. immigration system requires that petitioners become complicit in upholding the U.S. racial project and racial hierarchy as they negotiate their deservingness of a spousal visa with the state. Petitioners take charge of profiling couples for signs of fraud, drawing on socio-historically specific stereotypes of African American men, Arab men, and their family formation practices. Racialized judgments about foreign partners' reproduction, sexual agency, and breadwinning capabilities determine what is "genuine" and for whom. In doing so, they slot their partners into the U.S. racial hierarchy before they ever step foot in the country.

These findings center the petitioners as brokers of marriage migration policy, demonstrating the system's complexity beyond the "good" migrant couple/"bad" immigration official dichotomy. Further, these findings demonstrate the importance of investigating the

roles of self-help web forums in family reunification and immigration more broadly. Digital spaces provide resources for petitioning information and migrants' negotiation strategies. They serve as sites where borderwork occurs, not just inside the physical borders of the nation-state but also inside the virtual realm.

## Moral Line of Credit: Conceptualizing the Citizens' Role in the Moral Economy of Suspicion

The state has always regulated the family and intimate relationships[ii] for state-building and safeguarding the nation's and citizenry's integrity (Bonizzoni 2019; de Hart 2015), thereby controlling access to territory, resources, and membership (Macklin 2022). Before the twentieth century, western democracies used overtly racist and sexist laws tied to family and intimacy to prevent the immigration of non-white populations and limit citizenship rights to white men. Hence, the state granted citizenship based on one's race, gender, and country of origin, thus it was fixed (Gibney 2013). Once various disenfranchised groups pushed states to grant them citizenship rights, citizenship laws and immigration policies became seemingly universally-worded. However, t state bureaucrats and policymakers needed to reimagine new ways to police 'the family' to continue to control its interests. One such way they accomplished this was by using officials as "moral gatekeepers" (Wray 2006). State actors inside state bureaucracies use emotive evaluations and moral judgments of what constitutes proper family formations and practices to interpret policies and apply them to police and discipline citizenship and national belonging (D'Aoust 2018; Fassin 2005). Such evaluations shifted the citizen's lived citizenship experiences from a fixed, indisputable right to an "insubstantial privilege" (Gibney 2013) based on one's ability to conform with the state's moral expectations.

'Privilege-claiming' is when citizens try to perform and behave appropriately to obtain certain state goods and services. Whereas rights-claiming often involves citizens demanding their legally specified rights, such as due process, right to assemble, etc., defined by the law, privilege-claiming pursues conditional state benefits where officials must determine 'deservingness.' The welfare literature demonstrates that deservingness of who is and is not entitled to state benefits consist of moral determination based on several criteria, particularly control (i.e., ability to provide for oneself), attitude (i.e., good behavior, compliance), and identity (i.e., resembling 'us' or in-group) (Meuleman et al 2020; Reeskens and Van der Meer 2019; Van Oorschot 2000). The legal consciousness and welfare literature show how welfare recipients engage in privilege-claiming. Recipients demonstrate to the state that they are 'genuinely' deserving and 'in need' of financial assistance, which is usually closely scrutinized by bureaucrats using race, gender, and family norms (see Sarat 2006). Similarly, we can also extend privilege-claiming to the realm of immigration and family reunification.

The act of citizen privilege-claiming is part of an iterative, systemic process of governmentality intended to perpetuate the state's racial and gender projects of who belongs and who is a 'full' citizen. The literature on family reunification policy provides a comprehensive understanding of the state's participation. D'Aoust (2018) aptly describes this as a "moral economy of suspicion," which sets the interaction between state and non-state actors. The "moral" aspect of such an economy arises as family reunification policies and their interpretations draw on ideological discourses about "proper" family, which underpins the processes for rooting out marriage fraud (Bonjour and De Hart 2013, 2021). Officials engage in an "emotional" reading of artifacts, practices, and bodies to determine the adequacy of evidence for 'genuine relationships' that couples provide during the petitioning process (D'Aoust 2013: 334). As bureaucrats evaluate "technologies of love," or couples' specific relations, racialized pairings, and emotional displays through these ideological lenses of

family and belonging, they mark those who do not conform as suspicious; hence regulating immigration and citizenship of undesirable populations based on their understandings of "genuine love" (D'Aoust 2013, 2018).

However, such an 'economy' cannot thrive without r 'currency' to perpetuate it. This currency consists of the moral ideas surrounding family, citizenship, and belonging. Therefore, other actors must participate. The catalyst for such participation is the deliberate "holes" (Faist 2014; 43) in migration policy that the state expects "brokers" to fill through their interpretations (Odasso and Salcedo Robledo (2022; 174). Scholarship has shown that immigration lawyers (D'Aoust 2022) and NGOs (Odasso 2021a) serve as crucial brokers, but I suggest that the petitioners are also brokers as they traverse state scrutiny.  While many petitioners unfamiliar with the reunification process initially believe it is a simple bureaucratic one, they find their trust in the state over time called into question. (Odasso 2021b). As citizens begin to recognize that their marriages to non-citizens invokes the state's moral scrutiny, they start to evaluate what evidence that they must provide the state to overcome it. As a result, citizen-spouses find that their substantive privileges of citizenship diminished as the state subjects them to regulation, control, and scrutiny reserved for non-citizens even though their legal status remains intact (Macklin 2022). In other words, their legal status is fixed, but privilege is fluid and contextual.

To better understand the citizen's role within the moral economy of suspicion, I introduce the concept of a moral line of credit to analyze privilege-claiming strategies. Theoretically, citizenship grants citizens 'credit' in the moral bank, establishing a baseline of deservingness, albeit a varying one based on various social identities. Citizens, unlike non-citizens, have legitimate claims to their rights from the state (Macklin 2022) and can appeal to the state for certain benefits, like a spousal visa or welfare assistance. Whether the state will grant their requests depends upon their evaluations of deservingness. Moral scrutiny

begins when citizens appeal to the state for a good or service. Couples must demonstrate their "capital of authenticity" (Geoffrion 2017; 16) by negotiating the state's expectations of the family (Odasso 2020; 2021a). In other words, they must strategize to appear to 'naturally' align themselves as close as possible to the state's definition of the family ideal to demonstrate their deservingness. The moral line of credit is their bargaining position. Empirically, citizens can draw upon their character status assets, combinations of their race/ethnicity, gender identity, education level, and other social identities that emphasize whiteness, gender conformity, neoliberal values (Kofman 2018), and moral impression management. This negotiation is variable across citizens from different subpopulations and the nature and severity of what the state perceives as deviances or what I call their 'moral liabilities.' Those who cannot convince the state of their worthiness based on these strategies are considered 'overdrawn.'

The backdrop of such 'economies' is the historical socio-political hierarchies that have informed the nation-state's definition of belonging, nation, and citizenship. In the next section, I discuss the development and transformation of the U.S. racial hierarchy through state's policing of 'the family' to demonstrate where today's petitioners and officials determine what aspects of race, gender, class, and family are considered valuable moral 'currency' and 'liabilities.'

## The U.S. Racial Hierarchy, 'The Family,' and the Emergence of Privilege– Claiming

While whiteness has always been crucial for accessing social, political, and economic privileges (Roediger 1991), the U.S. racial hierarchy has always been gendered and classed. From the U.S. independence until the beginning of the civil rights movement in 1954 , a biracial order of whites and non-whites emerged (Bonilla Silva 2004) based on the nation's

reliance on the latter's exploitation, particularly on Black slavery, for its economic development (Feagin 2018). This economic reliance on non-white labor, enslavement, and oppression made whiteness a finite category dictated by skin pigmentation, cultural and religious differences, and national origins. Under the bi-racial order, Native Americans, Black people, Asians, Latinos, and even Irish and Italian immigrants (Ignatiev 1995) were classified as non-whites, which served as the basis for their disenfranchisement.

Gender played a crucial role in ensuring whiteness remained at the top of the U.S. racial hierarchy. Historically, the state constructs women as the purveyors of race purity and culture through their biological children (Hill Collins 2006). Therefore, the U.S. racial project concentrated full citizenship rights and national belonging strictly to white men with women and people of color into subordinate positions. To do this, lawmakers drew on hegemonic U.S. white, patriarchal family ideal as an essential ideological tool to upholding the racial project (Hill Collins 2006). The U.S. hegemonic family is a normative conceptualization, defined as a 'natural arrangement' based on heterosexual, married, white couples where men are heads of household and women are mothers and caretakers who are subordinate to their husbands (Longo 2018). By embedding 'the family' into policy and their interpretations, they serve the primary nation-building functions of border control and slotting different populations into the nation's internal racial hierarchy (Myrdahl 2010). Thus, until the 1967 Loving v. Virginia Supreme Court case that overturned anti-miscegenation laws, interracial relationships were illegal, particularly between white women and Black men. Under U.S. coverture laws that started in colonial times, women's citizenship was subsumed under their husbands, so they could neither marry foreign nationals without risking their citizenship status nor petition immigration on their behalf (Lee 2013). Based on perceived sexual immorality, under the Page Act of 1875, Chinese Exclusion Act of 1882, and Asian Exclusion Act of 1924, policymakers barred Chinese and Japanese groups from immigrating

and later implemented national quota limits on immigration for countries with large non-white populations (Lee 2013).

Thus, non-white immigrant groups also leveraged 'the family' as an ideological tool to claim whiteness. For example, many Arab Christians successfully drew on their religion, intermarriage with white, non-Arab women, socio-economic status, and good community standing to claim whiteness, allowing them to obtain citizenship rights and white designation by 1944 (Beydoun 2014; Orfalea 2005). Also, many non-white immigrant groups made decisions to stop sexual relationships and marriages with Black populations to stake their claims to whiteness, a requirement for citizenship (Wilkins 2004).

The U.S. racial hierarchy transformed into a loose tri-racial system comprised of whites' (i.e., traditional whites, European whites, assimilated white Latinos, etc.) at the top, the 'honorary whites' (i.e., Asian Americans, Middle Eastern Americans, light-skinned Latinos) in the middle, and 'collective black' (i.e., African Americans, Black Latinos, African immigrants, etc.) as the civil rights era of 1954 and a change in immigrant demographics and globalizing race relations emerged (Bonilla Silva 2004, 933). One's place in the U.S. racial hierarchy would now be dictated by skin pigmentation and social relationships such as intermarriage (Bonilla Silva 2004), less so by country of origin or culture.

The Immigration and Nationality Act of 1965 reflected the transformation by eliminating all racial, gender, and class restrictions on the petition process with seemingly neutral language. For example, the law now states, "to bring your spouse (husband or wife) to live in the United States as a Green Card holder (permanent resident), you must be either a U.S. citizen or Green-Card holder. Immigrant visas for immediate relatives of U.S. citizens are unlimited, so they are always available" (USCIS 2022a). Class requirements changed as well. A joint sponsor may assist if the petitioner's income does not meet the minimum income requirement (USCIS 2022b). Gender and race no longer dictated legal relationships. Instead,

the state now required proof of a "valid and subsisting" relationship, which defined marriage fraud as "individuals who knowingly enter into a marriage to evade any provision of the immigration laws" (Justice Department 2020).

Yet, the U.S. hegemonic family ideal continues to "haunt" (Turner 2015) the state's logic of who deserves full citizenship rights and who belongs. 'The family' builds on the colonial structure that required political power and socio-economic privileges remain concentrated in the hands of whites and white men, more specifically.  For example, political discourses constructed Black families as dysfunctional.  Government reports, like the Moynihan Report (1965), presented African American men as social problems by labeling them absent fathers, inadequate providers, and promiscuous partners. Stereotypes portrayed them as duplicitous, lazy men who used their hypersexuality on unsuspecting women to obtain economic gains that they have no right to, deeming them irresponsible, immature, and unable to handle the responsibility of full citizenship rights (Hill Collins 2005; 162). Even though laws on interracial relationships changed in 1967, working-class white women in relationships with Black men found their rights and privileges increasingly marginalized due to their intimate association (Hill Collins 2005).

Certain minority groups could still leverage 'the family' as an ideological tool to claim or safeguard their place in the U.S. racial hierarchy, and others could not. For example, after September 11th, the "War on Terror" campaign demonized Arab populations, particularly Muslims, as "political pariahs and national security threats," leading to the erosion of Arab access to privileges and power associated with "legal" whiteness (Beydoun 2014). Despite this erosion, many Arab Americans continue to embrace their honorary white social status through their social relationships, such as intermarriage with non-Arabs and performing 'proper' family norms. More than 80 percent of U.S.-born Arab-Americans are married to non-Arab women (Orfalea 2005). Research shows that non-Arab women married to

immigrant Arab men believe that their husbands took the role of breadwinner, father, and head of household more seriously than today's American men (Longo 2018). Hence, many Arabs can maintain racial ambiguity in ways that Black populations cannot, in part through their performance of the U.S. hegemonic family ideal.

From the history of the U.S. racial project, a systemic process has emerged where groups began to draw on different social identities and normative performance of family to claim whiteness and, to that end, citizenship rights. Today, however, citizenship is not tied to race, class, and gender anymore, but the ability to get privileges from the state is. As we shall see, U.S. petitioners lean on understanding these social identities and the U.S. hegemonic family performances to claim visas for their foreign partners. In doing so, they become complicit in upholding the state's interests and engage in nation-building.

## Methods

This study is based on an online ethnography and content analysis of "Immigration Pathways," one of the largest U.S. immigration self-help websites; as of 2018, it boasted 113,000+ members and 2.2 million posts. Forums are sub-divided by regions, and visa types and are entirely public. However, individuals must become members and use a consistent pseudonym to post, producing online personalities and reputations. Thus, many forum members consider individual peers regional immigration "lay experts," believing their advice to be highly accurate. Members cultivate expert status through collective experiences, observations, anecdotal evidence, and membership longevity. They also base advice on their perception of what immigration officials will do. Members optionally disclose unverifiable demographic information. However, forum members report that they are mainly white and Black, heterosexual U.S. citizen women, and men in transnational relationships. Smaller groups of foreign nationals and bystanders also are present.

Initially, I was ethnographically immersed for two and a half years, strictly as an observer.  I chose Immigration Pathways because it was one of the longest continuous websites under the same administrators.  Initially, I was ethnographically immersed for two and a half years, strictly as an observer.  I did not interact with participants. These observations afforded me insights into the cultural dynamics of the community, the important topics of conversations, and the cultivation of member status described above.

For this study, I analyze conversation threads from two regional forums on Immigration Pathways. One is the Sub-Saharan Africa forum for U.S. women married to Black men of African origin. The other is the Middle East/North Africa forum for U.S. women married to Arab men living in the region. Both regions are considered high risk for marriage fraud and thus endure greater government scrutiny. Each forum consists of a specific racial demographic of foreign partners and racially distinctive criteria for labeling genuine or fraudulent relationships. Although these online discussions cannot reveal what actual immigration officers consider fraudulent, they depict members' experiences with immigration officials and their understanding of genuine marriages for immigration purposes.

The MENA-forum contained 9,039 separate conversation threads, while the SSA-forum had 3,345. I compiled a purposive sub-sample of 668 MENA and 319 SSA conversation threads that consist of the term red flag and have five to 30 replies from three or more people who responded to a post. These threads generally contained debates over why additional scrutiny was occurring among particular petitioners. I open-coded these to classify different themes. Then I iteratively constructed a closed coding schema for analytic themes such as race, gender, age differences, fertility, family, and economic support.  In this way, I could glean how members constructed their understandings of family.

The sample is not random. It represents people with immigration concerns who self-select into the site. It over-represents English speakers and probably over-represents people

who anticipate difficulty obtaining a visa. LGTBQ people are only marginally present in the forum and not part of my analysis. However, online comments to peers are less prone to reactivity bias than face-to-face interviewing since members naturally produced discourse in the problem-solving process rather than generated in response to researcher questions (e.g., McMillian Cottom, Daniels, and Gregory 2017; Longo 2018).

The first three sections of the findings illustrate how petitioners negotiate the moral economy of suspicion using their moral line of credit. First, I show how petitioners interpret state policies to determine what the state is morally scrutinizing. Second, I show how petitioners consider ways to package their evidence to demonstrate their genuineness. By understanding what the state is looking for, they compile their evidence packages to showcase their moral assets and mitigate the red flags in their relationships. Third, I show how petitioners negotiate problems particular to their cases using their moral line of credit. In the last section, I demonstrate how the more extensive process of privilege claiming requires petitioners to be complicit in upholding the racial project. As they engage in this process, they slot their partners into the racial hierarchy before they ever step foot in the country.

## Findings

### Defining the state's 'costs' for 'earning' a visa

Couples must prepare to showcase their moral assets and mitigate their moral liabilities to obtain the privilege of visa approval. However, petitioners cannot do this until they know what the state expects and where they meet or fall short of these expectations. Many members encounter a steep learning curve due to their inexperience with the state and rely on more seasoned members' journeys. They naively believe the state will not challenge their petition (Odasso 2022) as they conflate the privilege of approval with their rights as a citizen (Bloch 2021).

For example, LilSaraSoozy, a white, 20-something recent college graduate from a well-recognized university, is a new member engaged to a young Moroccan man she met during a college study abroad. She asks whether eight months out from her fiance's initial petition is too early to plan an Alaskan cruise wedding. "Seriously I don't expect there to be any problems. We are probably the most textbook case! LOL!" Members quickly disabuse her of these assumptions. Goldberry1313 immediately replies with the quote at the opening of this article. 1MomoNmeg follows up, "Agreed...you know the old saying---Asking ain't getting." Thus, early on, forum members learn that their citizenship status will only get one so far (Odasso 2022; Macklin 2022). Peers much further along or who have completed the process explain that couples must show that they are genuine and can explain away any red flags to "earn" the visa.

Members draw on the U.S. hegemonic family ideal to fill in the "holes" of U.S. immigration policies (Odasso and Robledo 2022). They speculate what the state is looking for in a "valid and subsisting relationship" or how it defines indicators of marriage fraud (i.e., "red flags") like "age differences," "vast cultural differences," "requests for money," and "length of courtship" based on other's previous experiences. While the racialized, gendered, and classed understandings of the family shape the definitions of "red flags," the couple's race in particular shapes the severity of "red flags" across and within the forums.

For example, 1LUV4, married to a Somali man, posts that their case has been subject to additional investigations for potential fraud for over eight months. As forum peers send supportive messages, NigerianLove, a white woman who successfully petitioned for her Black Nigerian husband, asks 1LUV4 if she is white. Another new member curiously inquires, "if I may ask,does it matter if the petitioner is black or white?" Forum members agree that most often, it is white women-petitioners and their Black partners who report the most considerable barriers to getting their petitions approved. Clarification follows:

NigerianLove: "My husband was asked [at his interview] why he would marry a white woman when he could have any black woman he wanted...Sadly, YES the racial difference plays a big part in the visa process, as well as gender and prejudice in regards to age difference."

WSHngONaSTR: "Looking at the pattern of denials to white women petitioning for black spouses shows me the embassy has got some rather outdated ideas of race."

The issue is not necessarily about blackness. Instead, it is about interraciality and gender. Members agree that the state perceives red flags for white petitioners married to Black men as more severe. For example, when someone asks about a nine-year age difference between her and her Nigerian partner, Jan&Simo, an African American woman who is considered a lay expert, responds:

"It's never just age...Different races. Divorced. Children from other relationships or marriage matter…but, if you look at the pattern of issues during the interview it is mostly mixed-race couples with an older (white) woman [that get denied]."

These conversations demonstrate the subforum's consensus that immigration officials evaluate the integrity of couples with Black immigrant husbands using an extremely high bar, particularly those married to white women. SSA-forum members are keenly aware of how the state's expectations for racial purity can affect their cases. African American women who marry Black African men report that they do not experience the same racial scrutiny. They are considered racially matched since the U.S. racial hierarchy rests on social relationships such as marriage and skin pigmentation (Bonilla Silva 2004). Unlike European immigration officials' emphasis on ethnicization (Wray 2022), cultural differences and community take a backseat to racial differences.

Similarly, the Middle East/North African-forum women agree that the state's moral scrutiny is grounded in racial purity and conformity to the hegemonic U.S. family ideal. But,

MENA-forum women are unsure exactly how racial purity impacts immigration officials' rulings on their petitions because they are uncertain how to classify Arab men racially. For instance, many members openly describe Arabs as racially different, regardless of religion, from whites: "I think Arab is a race...[Immigration officers] seem to also", says BETrTHnEvr.  As one member posted, "...well, I would not say they are 'different,' but they aren't the same." Despite ascribing racial otherness, none considered themselves in an "interracial relationship." None seem concerned about the impact of a couple's racial composition on their cases or reported that immigration officials questioned Arab husbands about why they would marry white (non-Arab) women as seen in the SSA-forum.

However, race does become a factor when Arab partners appear to violate gendered and classed family norms, such as couples where the women petitioners were older. For example, Msierforever, a 40-year-old white woman, asks whether immigration officials will consider her age difference with her 22-year-old-Egyptian husband a red flag.  Given the age difference, members racialize Arab men by drawing on 'cultural difference,' particularly "their" family formation practices as a proxy for race.  Once racialized, they portray them as dangerous, racialized others:

> MissMLK28: "Most MENA-cultures are very paternalistic and center life around the family unit. It's uncommon for a young man to accept marrying a partner who cannot give him children or is edging up toward menopause."

Another member replies, "Consulate officers KNOW that doesn't fly in their culture." SandnSun, who has successfully obtained a visa for her Egyptian husband, and who is very respected on the board, follows:

> "That's right...they are going to think you are a 'starter wife.' A lot of time, [Arab men married to older women] marry you so they can get established and then bring one of their own [women] here."

Like the SSA-forum, MENA-forum members try to determine what the state is looking for and evaluate their petitions using the hegemonic family. Race, gender, and later, we will see class dimensions of the family matter. Yet, the MENA-forum demonstrates precisely how the petitioners also contribute to these criteria. As members put on their 'consulate hats' to interpret policies, they also evaluate peers' relationships based on what they feel a 'proper' family looks like. Then they apply those logics to others' relationships. When couples cannot conform to these expectations, then Arab husbands, as "traditional," "backward," or "old-fashioned" others, are treated as racial infiltrators, indicated by the racialized buzzwords like "their women" or "their kind." Members warn one another about "bezness" (Scheel 2017),  or the phenomenon of Arab men engaging in sexual relationships with non-Arab citizens strictly for immigration purposes, which not only casts doubt on his integrity but implicitly accuses the petitioner of being duped and calls into question her ability to safeguard the nation-state. However, when these partners conform to family norms, members do not mention Arab husbands' cultural family values or accusations of "bezness" (Scheel 2017), even when Arabs marry non-Arab or non-Muslim women or have pre-marital sex. Therefore, to remain racially ambiguous, Arab partners must want biological children and earn money for their family, those who do not are said to be hiding their true intentions.

Ultimately, petitioners in both forums must determine the standards by which the state will judge them to strategize how to draw on their moral line of credit. To do this, they will need to garner their "capital of authenticity" (Geoffrion 2017) via performing proper family and showcasing their family's race, gender, and class dimensions to overcome the state's moral scrutiny. Through these conversations, participants identify the contours of the racial project in all of the project's complexity. They demonstrate that the impact of race on "earning" a visa is nuanced. It reflects the tri-racial hierarchy with Arab partners constructed

as "honorary whites" who have the potential to garner some white privilege and Black men as "collective Blacks" who do not have the same opportunities to do so (Bonilla Silva 2004).

**Drawing on the Moral Line of Credit in a Moral Economy of Suspicion**

Petitioners must showcase their deservingness through tangible artifacts (D'Aoust 2014) that demonstrate genuine relationships. The U.S. immigration services provide a very sterile list  of some examples of possible evidence, such as "birth certificates," "joint bank statements," "photographs," and "phone records."  Petitioners may also choose to write a narrative or have family and friends submit letters of character on the couple's behalf. New members are eager to put together their evidence packages. Still, seasoned members remind them that they must put themselves into the consulate officer's shoes and decide what to include and omit.   With this in mind, members analyze another's relationships and weigh the strength of the evidence using the hegemonic U.S. family ideal. In doing so, they keep intact the superiority of the nation over the private matter of marriage that gets reconfirmed in the process and leaves intact the need to identify potential swindlers and suspect 'others.'

Questions arise about whether having children together before petitioning or being pregnant constitutes proof. For instance, in the SSA-forum, Sandra&Luke has been in a long-term relationship with her Nigerian fiancé, and they have a two-year-old daughter together. Because U.S. immigration once denied him a U.S. student visa, she asks if their child will help demonstrate proof of genuineness. Omba18, an African American woman who successfully petitioned for her Nigerian husband four years before replies:

"Remember that in [the immigration officials'] minds a child is proof of sex, not love. Are you the same race? That will normally help. You need to show that you are a couple and you and the child are not an excuse for a green card."

Similarly, JessaCLA, a white U.S. woman currently living in Kenya with her fiancé of six years and their child, has been subjected to a lengthy investigation for marriage fraud. She

angrily asks why it takes so long to complete the investigation and what kind of evidence might speed the process up. Members explain:

> *StandingProud1234:* "They are probably wondering why you did not marry and file after having a 4-year-old daughter and 6-year relationship."
>
> *HalahahB:* "I agree with the red-flag, I think that they [ask themselves] why did you have a child and have not been married yet after six years."

In the SSA-forum, members use the discursive narratives about African-American men as hustlers and poor fathers (Hill Collins 2006; Moynihan 1965) to evaluate evidence. Because race, not ethnicity or cultural differences, is more salient in the U.S., the same racialized and sexualized rationales about African American men also underpin evidence discussions for Black immigrant men. Therefore, the presence of a child or pregnancy does not lower moral scrutiny. Instead, they suggest that it indicates that SSA men will "hustle" (Hill Collins 2006) U.S. citizens into having biological children simply for a visa.   This seems to make it far more challenging for SSA-couples to demonstrate their performance of 'family' and show their worthiness.

In the MENA-forum, the question of children arises as well. However, historically, some Arab men have successfully aligned themselves with the U.S. hegemonic family. As such, MENA-forum members agree that children are not proof of genuine relationships, but they are not considered a suspicious red flag, unlike the SSA-forum. For example, several members reply when StaceyYosef, who is petitioning for her Egyptian husband, wonders whether disclosing that she is newly pregnant will expedite the process or serve as a red flag.

> *DellaS467:* "From what I understand having a baby with a USC married or NOT will not affect your visa process in any way. You are the USC, you can happily have your baby and carry on with the process you wish. A baby does not grant grounds to expedite your case nor does it affect it in a negative way."

*WaitinginNY:* "Your longest wait will be the USCIS leg, and pregnancy won't speed

them up. As for proof of ongoing relationship, no a pregnancy alone doesn't prove

there's an ongoing relationship."

*MollyMoeMo:* "I got pregnant in Morocco, and we were not married yet (oh the

horror!) Nobody cares."

MENA-forum members agree that children are not red flags, although they are not

proof of genuine relationships. Suggestions that husbands with U.S. citizen children may be

using these ties to scam their way into the United States are notably absent. It is due, in part,

to the stereotypes of absentee fathers and hustling historically aimed at Black men, not Arab

men. The presence of biological children in MENA relationships even before marriage aligns

with the racialized and gendered U.S. family ideal. It neither helps nor hinders a couple's

case. Hence, the concern about "bezness," so pervasive in the European discourse, is less

prevalent.

But this does not mean that MENA relationships are immune to moral liabilities.

Gender, race, and class dynamics also matter, even if MENA couples seemingly struggle less

to demonstrate 'proper' family. Those who do not conform to gendered class aspects of the

'family' have their Arab partners racialized as dangerous 'others.' Therefore, petitioners

cannot assume that immigration officials will take just any proof at face value.

For example, some new members ask if sending money can be considered proof of a

genuine relationship since married U.S. couples often comingle finances, and the state lists

"bank statements" as evidence. "Scammers invent all sorts of excuses for a USC

fiancée/spouse to send money to help at very inflated prices via local standards," wrote

Farmergirl15. Foreverhisgirl continued, "With all due respect, I think that if the guy asks you

to send money, then it is time to re-think the man." The exchanges continue regarding

sending money to Arab partners. Towards the end of this conversation thread, one new

member posts that her experience in her husband's country of Morocco does not align with the warnings she has received about men scamming women out of money. "Morocco is full of genuinely kind people, and life is simple," she notes. HappyGal34 scoffs, "I started seeing the young prostitutes everywhere--fathers and brothers pimping out their daughters and sisters so the family could eat."

Failure to draw on the normative expectations of gendered and classed family expectations distanced them from whiteness. It is not whether husbands have good-paying jobs but whether husbands take money from their wives that make the difference. This is not a coincidence. The American construction of marriage, love, and nationhood aligns with disinterested love (Honig in D'Aoust 2022; 125). There is an expectation that there is no material gain when a foreign national marries a U.S. citizen. Such expectations are gendered, however. MENA-forum petitioners must also carefully strategize when drawing on their moral line of credit.

Because privilege is "insubstantial" (Gibney 2013), what is considered an 'asset' in one situation may be a 'liability' in another. Therefore, how petitioners draw on their moral line of credit involves more than performing family norms. One must also know when to emphasize their 'character status assets' or mute them in conjunction with their performative work.

For example, in the SSA-forum, @ishadreems2, a white woman with a graduate psychology degree asks if she should discuss in their narrative how her husband, who holds a high school diploma, plans to get his college degree. She also wants to talk about how they plan to run a joint counseling practice after his licensure. "He loves helping people and already works in community services outside of Lagos, and a counseling practice is our dream!" Members discourage this tactic, suggesting she mute her class privilege:

*MsinNY: "*The CO might wonder what a guy with a high school diploma is doing with a woman with your credentials.  Might be a red flag,"

*Jan&Simo:* "Yeah, agreed.  Would your parents or friends be willing to write a notarized letter stating that he will get his counseling license and be a hard worker?  It puts the focus on him--leave your 'dreams' (and you) out of it."

Class is becoming increasingly important and may, in some ways, trump race in the nation-building project (Kofman 2018). Class privilege and higher socio-economic status open up so many opportunities for citizens, especially those from the racial majority. So, it is understandable that some petitioners assume their high-status job and higher levels of education will negate intense moral scrutiny (Odasso 2022; Bloch 2021). But, such assumptions are a mistake because trying to demonstrate that one's partner will be a 'good neoliberal citizen' is a gendered and racialized negotiation. Consequently, it is crucial to draw on the correct character status assets. For men, hypogamy is acceptable, but not for women. The gendered mismatch between education levels in @ishadreems2's case potentially creates a moral liability, compounded by their interracial relationship. To showcase her everyday privilege of being a white, educated woman in the United States in her narrative may not serve as a character asset in this circumstance. It may reflect poorly on the Black immigrant man with less education because of the narrative about Black men as "hustlers."

Moral evaluations are not one-sided. Although immigration officers' moral evaluations of "technologies of love"  (D'Aoust 2018; 2013) are a vital part of the moral economy of suspicion, the moral line of credit serves as the other major component. Petitioners take the curation of these items very seriously. They discuss how much evidence should be submitted, what to offer and how to present it. They must perfect their curations of evidence based on the feedback from others' encounters with immigration officials and their

understandings of the hegemonic family. In doing so, petitioners construct their subjectivities

and define what relationships are 'genuine' enough for marriage migration.

**How petitioners negotiate being potentially 'overdrawn'**

Some cannot put forward enough convincing evidence of a "valid and subsisting"

relationship during the initial petition, visa application, and the interview stages of the

immigration process. Consequently, a subset of forum members experience problems with

their petitions, such as denial, further fraud investigations, or requests for more evidence.

These petitioners realize that they are potentially "overdrawn in the privilege claiming

process." In other words, with too many "moral liabilities" and not enough "moral assets,"

they cannot counter the moral scrutiny at some point during the process. Some recommend

"rights-claiming" tactics, such as hiring a lawyer, contacting congresspersons, or even

reaching out to the press to apply pressure. However, most petitioners maintain that this is

ineffective and that they should continue to use on their character status assets and

performance of 'family' to overcome the 'overdraw.'

In the SSA-forum, marriage fraud field investigations and three-way DNA tests are

common, making couples more prone to being "overdrawn." For example, WeWait, an

African-American petitioner who had her child on U.S. soil while her husband remains in

Ghana, laments that officials put him into Administrative Processing, the term for post-

interview fraud investigations, at his consulate interview:

> "When my husband went for his interview, the consulate officer said that he has to
>
> (delay) his approval based upon the results of a 3-way-DNA test. The test is to prove
>
> that our child is paternally his and maternally mine."

She continues to lament that she tried contacting her local representative to intercede

on her behalf, but they said they could not help. When one of the newer members suggests

that she get a lawyer, other more experienced members balk: "That's a waste of time because that is not a violation of your rights. They are going to say that everyone with kids has to submit to the DNA tests. That's why the rep can't help," NigeriaOrBust replies. More peers recommend that the couple submit more candid photos of together playing with their child to the consulate officer to show they are a family unit. Halahah explains:

"It's important to show that (your partner) is involved actively in the child's life at every stage. Pictures are really useful for that. Send as many as you can. The [consulate officer] needs to see that this child isn't just some way to get a green card."

In some cases, being 'overdrawn' is not the couple's fault but through situations beyond their control. Petitioners must quickly choose their most favorable character assets to remedy the issue, which includes their citizenship status. This is different than rights claiming. Instead of exercising their rights through legal channels or protest, petitioners as citizens have specific ways to showcase their social status demonstrate norm comportment in ways that non-citizens cannot, thereby exerting their influence over the process.

For example, consulate officials will reportedly initiate extra background checks, including field investigations, which entail surveilling partners abroad and interrogating neighbors, family, and friends in their hometowns. MonicaNCooper, a white woman petitioner, posts that she is apprehensive after receiving a text message from her fiancé. Her fiancé notified her that an unidentified consulate officer came to his residence and tried to question his mother. He was still at work, and his mother had refused to speak to the investigator because the officer would not disclose who they were or what they wanted. They just said that they wanted to ask questions about the couple. Several longtime members quickly respond that their partners have had similar experiences and advise them how to negotiate their botched field investigations successfully.

*Lexie&Bassir8:* "I was told by others to contact the field investigator, since she gave me her email address and say that my mother-in-law called me immediately (after realizing her error) and was concerned that her hesitancy to answer questions from a stranger might affect her son's case."

*Ingrid1984:* "Yeah, [Another member] said that I should send framed pictures of myself and my family from our wedding to my husband's mother's house long before the consulate interview takes place. That way before the investigation even starts, they are on display."

MonicaNCooper later wrote that she had left a detailed voicemail message along those lines and sent a detailed email to the consulate. In these communications, she explained that her mother-in-law called her very upset to say that she realized her error, but at the time, she was protecting her son from a stranger asking questions. She states that she never received a reply and reports that the consulate officer never returned to investigate his family a second time. Six months later, she posts excitedly, "VISA IN HAND! THANK YOU ALL FOR YOUR WISDOM AND SUPPORT!! (Cooper) will be here in three weeks!"

The data cannot tell us just how impactful this interaction was on the consulate officer's overall decision to approve. Still, it does reveal how citizens draw on their moral line of credit to negotiate deservingness and claim privilege and how this upholds the U.S. racial project. In this case, the petitioner's citizenship status served as a character status asset as she could claim the privilege of directly contacting the field investigator. Immigrant beneficiaries do not have the moral line of credit to do so themselves. She could perform "intimate citizenship, where citizens express what citizenship is and should be, by mobilizing intersecting conceptions of intimacy and of belonging" (Bonjour and de Hart 2021, 1) based on her ability to contact the investigator (Odasso 2020). Telling the investigator, for instance, that the citizen's mother-in-law called her rather than the petitioner's husband signals that she

has ongoing direct communication with influential members of his family. In justifying the refusal of their mothers-in-law to answer questions about their sons, petitioners use the normative expectation of motherhood that a woman's first instinct is to protect her children.

Unlike the SSA-forum, very few petitioners report that their partners receive field investigations, and there were no reports of three-way DNA tests. Instead, MENA-forum couples often have the most potential for 'overdraw' at the consulate interview. To ensure they demonstrate that their partners can do family (and whiteness), they package additional evidence for the interview and coach their partners on appropriate answers during questioning. They must be sure that consulate officers do not perceive them as dangerous "others." Consequently, they attempt to minimize their husband's racializing ties to religion and Islamic culture while accentuating their conformity to the family ideal. For instance, SandnSun advises another member whose Moroccan husband is going to his interview that he purchases a suit and groom himself appropriately:

> "For God sake, tell him not to shave his beard completely off the night before he goes for his interview. Dead giveaway that he has had the 'jihadi' beard that scares the shit out of consulate officers."

Likewise, members often remind one another to tell their husbands not to say "I hate Al Qaida" to the consulate officer because it makes them appear more suspicious. In a similar thread, one seasoned member, Goldberry1313, recounts how husbands need constant coaching because every decision, regardless of how trivial, is being scrutinized with racialized suspicion:

> "The Moroccan consulate is always going to ask your spouse if they want the interview in English or in Arabic. If English is YOUR only language then he should want the interview in English. C.O.s will think that he cannot communicate with you and is trying to scam you. They aren't just trying to be helpful to your partner!"

Photos are one of the "technologies of love" (D'Aoust 2013) that consulate officers evaluate. So, they must be carefully chosen and even crafted, making petitioners the "engineers" of such technologies. When members discuss the importance of photos that spouses can show at the interview, they urge them to pay attention to the body language in the pictures. Each image must show appropriate emotion typical of an 'American' couple. In many MENA countries, cultural norms dictate that you do not smile or show gratuitous physical contact in pictures. For example, a general rule of thumb is the "toes in" feature. Members will advise newcomers to submit pictures where the couple turns their bodies in towards each other, so the couple's feet are pointing to the others. As MissMLK28 puts it, ,"simply standing next to each other ESPECIALLY if you are hugging each other shows a lack of affection on the hubby's part. [The Consulate officers] will think he is duping you."

These conversations show that moral scrutiny differs across the SSA- and MENA-forums. SSA-forum couples report being overdrawn far more often in the SSA-forum. In contrast, MENA-forum couples find that the potential for being overdrawn is most likely to occur at the interview phase. It is up to the petitioners  to draw on different combinations of character status assets, shaping how petitioners strategize how to best draw on their moral line of credit.

**'Social Problems' and 'National Security Risks':  How privilege claiming reproduces systemic racial inequalities**

Family migration policies and regulations are exercises in state-building and nation-building. As citizens try to secure their own family, they must partake in such practices by disciplining themselves to align with the state's ideal of what a nation, a citizen, and a family should look like. Couples must negotiate by the game's rules, no matter how inequitable, to claim the privilege of approval. Lawyers, NGOs, and other volunteer organizations can serve

as "brokers" of family reunification laws and regulations as they attempt to fill in the substantive meanings of the policies (Odasso and Robeldo 2022; D'Aoust 2022). But, for petitioners, the system demands that they become brokers, whether they would like to or not.

For example, racism is present on the SSA board, but there is also pushback against the overtly racist characterization of Black husbands. Whether a petitioner supports the racist ideas that underpin the hegemonic U.S. family or finds it appalling, they all must meet the state's expectations for a "valid and subsisting" relationship. They must construct their deservingness based on the racialized, gendered, and classed ideas of 'proper' family, even if it disenfranchises them in the long term. Take the new member who asks, "Is it a race thing or an African thing? Do the white Africans go through as much harassment as black Africans?" others chime in:

> *MsinNY:* "I think it's more a financial or class bias than a race bias. [Consulate
> Officers] that I've spoken with are acutely aware of the way so many people in poor
> countries see a visa to the U.S. as winning the lottery."
>
> *By-ding-TYME:* "Sad thing is lots of people want to get to the U.S. so they can
> eventually go on welfare."

The backlash is immediate. "Umm...is there even welfare in this country anymore? Most come here for love and family," Omba18, an African American woman, counters. In another conversation thread, StarsRshining4US tries to blame the government's "old" and "outdated" views about race on Black men. She says, "part of the problems blacks have isn't as much racism as it is the image some (Black men) have created.... Look at all the drugs and violence that are glorified as art today--The disrespect of womanhood, the law, and life."

> *RamonILUVU* (female, African American petitioner): "Black culture you experience
> on tv, many films, music videos, do NOT represent African Americans. Did
> "Roseanne" represent all white people in America?...it is the negative side of Black

America that is marketed and promoted, undoubtedly skewed as lower-class and violent."

*MoniQ22* (African-American, female petitioner): "I feel it's about a black racist hatred...like ,'why are we shipping all these black men to America, we don't want them.'"

Comments that resist racism challenge peers' perspectives, not necessarily the moral parameters of the privilege-claiming process. African American citizen women, like Omba18, call out racism, but earlier, when discussing whether a child was considered proof of a 'genuine' relationship, Omba18 assessed her peers for normative violations of the family that questions the motives of Black men who have children out of wedlock. She used the same racialized socio-historical discourses that construct Black men as hypersexual, absent fathers, and "hustlers" (Hill-Collins 2006) to contextualize her advice.

In other words, members label Black immigrant men as 'social problems,' using the same stereotypes that marginalize African American men. Marking them 'social problems' makes it harder for SSA petitioners to purport genuineness. However, if they successfully obtain the visa, they have already slotted their partners into the "collective Black" (Bonilla Silva 2004) tier of the U.S. racial hierarchy as stereotypical African American men in the making.   As Black women understand how this portrayal of Black men has underpinned their marginalization (and their own), it is doubtful that they agree with or desire to uphold such characterizations. But, if citizen-spouses do not play along, they risk prolonged separation, financial hardship, and even self-imposed exile. They have little choice.

In part, the stereotypes about Arab men come directly from the government's concerns and its political dialogue on the so-called "War on Terror."  MENA-forum petitioners also must be "brokers" whether they want to or not.  Petitioners must keep in mind

that partners who do not conform to intersectionally gendered expectations of family risk being labeled 'national security threats' based on the post-9/11 era.

For example, when Americanwoman869, age 40, asks if the state will be concerned about the age difference with her 22-year-old Egyptian husband, members suggest that this will be problematic. HisGirl132 seems to think there may be a chance. She replies, "a woman 18yrs a guy's senior is not too bad if she can still procreate." But as the thread unfolds, some start to frame the issue as a matter of national security.

*VioletsRBluL*: "Males from certain "T" countries are required to undergo [more investigation]."

*ClearSAALNG:* "Persons coming from T countries (Terrorist Countries) seem to have. more security checks... Common names, your nationality, religion, education, relationship of applicant, travels and job can all be factors."

Petitioners recognize that the state will view Arab partners as potential national security threats. Therefore, they must frame their warnings to those with red flag relationships in these terms, whether they want to or not. However, petitioners will also use 'national security threat' discourse to slot Arab men into the U.S. racial hierarchy intentionally. MENA-forum women who primarily identify as white actively use the U.S. government's so-called War on Terror narrative to racialize Arab beneficiaries who challenge or question their complicity in perpetuating inequality as a way to keep them in their place.

For example, LuxorGuy, an Egyptian male beneficiary, writes that he cannot understand why so many American women on the sub-forum accuse Arab men of being terrorists when married to Arabs. "...that is just a government excuse because they have problems with Arabs and want to treat us bad," he states. Petitioners angrily respond:

*MindySOFLA:* "heaven forbid if a country try and do additional check on countries where more than 99% of the terrorists with ill intentions for the U.S. come

from...Being an Arab for you is a problem not because the U.S. is trying to stop you from coming in but because most of your kin with similar names are trying to come in with all kinds of destructive intentions instead of calling the people [who are] trying to stop another 9/11 from happening as 'Racists'… maybe you can have the balls to tell those terrorists to 'Stop' so you can have a more normal life just like the rest of us".

*SarahMO1101:* "I would rather they take the time up front to single out the threats to our safety. It is amazing that you will never hear Arabs denounce those who give their race and culture a bad name. It's much easier to whine about racism".

Unlike SSA-forum petitioners who vocalized their rebukes of racism openly and often without backlash, Arab beneficiaries do not receive the same respect. Petitioner members actively apply the status of national security threats and terrorists to those on the board who do not stay in their 'place.' It serves as a form of rebuke to non-citizen beneficiary men for speaking such stark criticism about the U.S. racial hierarchy. In other words, MENA-forum white petitioner women see these men as outsiders having no right to critique the U.S. racial project. It serves as a reminder of who may legitimately challenge power and who should not.

## Conclusion

Family reunification scholars (Myrdalh 2010; Bonjour and De Hart 2021) argue that government policies for the detection and prevention of marriage fraud constitute racial projects, despite their seeming race and gender-neutral language. Previous scholarship demonstrates that bureaucrats and state actors engage in moral gatekeeping as each evaluates who is deserving of immigration using their understandings of what constitutes a "proper" family (Bonjour and de Hart 2013; D'Aoust 2013; Wray 2006). As such, non-state actors such as lawyers (D'Aoust 2022) and NGOs (Odasso 2022) must serve as "brokers" who will

provide context and meaning to these policies, so petitioners can successfully reunify with their partners.

How petitioners contribute as brokers of immigration policy to the racial project remains undertheorized. This study fills these gaps by investigating how the petitioners' strategies to achieve spousal reunification contribute to the racial project. I show that petitioners connect generic relationship criteria and fraud warnings in U.S. immigration policy with intersectionally organized racial ideologies, behaviors, and expectations about the U.S. hegemonic family to evaluate what evidence constitutes 'proof' of a genuine relationship and for whom.

There is an implicit dichotomy between the "good" couple and the "bad" immigration official within the literature. This study troubles this dichotomy while giving priority to the petitioners' voices. Drawing on the welfare and legal consciousness literature (e.g., Sarat 2006), I show that petitioners must engage in 'privilege-claiming' as obtaining a visa is not a right. They must demonstrate to state officials that they are 'deserving' of a visa by aligning themselves with the state's idea of family. I introduce the conceptual framework of a moral line of credit to illustrate the systemic nature of privilege-claiming, where the state and petitioner play essential parts.

Couples must negotiate the state's demands, no matter how inequitable, to claim the privilege of approval. Otherwise, they will potentially experience denial, financial hardship, and prolonged separation. Therefore, petitioners interpret the family reunification requirements by putting on their "consulate hats" and using the state's definition of family to provide the appropriate feedback to immigration officials.

Petitioners must be complicit in upholding the U.S. racialized hierarchy. They forge their own subjectivities of citizen and nation. SSA-forum petitioners apply socio-historic political race discourses about African American men that construct them as social problems,

absent fathers, and "hustlers" (Moynihan Report 1965; Collins 2005) to discuss marriage fraud among SSA-region men. This implies that Black men, in general, lack the integrity and maturity to hold U.S. citizenship or green cards (Collins 2005). MENA-forum petitioners treat MENA-region men who cannot perform whiteness through their role in "the family" as national security threats, characteristic of the post 9/11 discourse about Arab men. Each is seeking the best way to demonstrate their deservingness by showcasing how well they align with the white, patriarchal contours of "the family."

As a result of this process, citizens perform borderwork, and the digital spaces are places where borders are made. Online forums serve as publics where fierce debates about family, race, gender, class, and nation occur. Digital publics are unique because they bring together populations of people, such as marriage migrant couples, diverse refugee groups, and other migrants, together in numbers not necessarily found in physically situated places. Their conversations remain memorialized in digital archives that can stretch back for years. Future immigration scholarship should dedicate their time and attention to the digital space, treating it as a new avenue of theoretical inquiry of immigration and its politics.

# Endnotes

i.  Website and user names are pseudonyms

ii. Please note that the literature on family reunification is particularly large. There are comprehensive, overlapping scholarships on migrant parent-child, migrant children-parent, and spousal reunification. Moreover, there are different conceptual frameworks to examine family reunification, such as through socio-legal dimensions, kinship entanglements, and so on. To address all of these areas goes beyond the scope of this paper, but the author wishes to acknowledge the sizable corpus of work in this area.

## Acknowledgements

It takes an engaging community doing incredible amounts of invisible labor to bring forth a piece to publication. The author wants to thank the colleagues, reviewers, friends, and mentors across the globe who assisted me with their intellectual feedback and encouragement with this work. I am thankful to Drs. Saskia Bonjour, Anne-Marie D'Aoust, Paola Bonizzoni, Betty de Hart, Darshan Vigneswaran, Evelyn Ersanilli, Myra Marx Ferree, Gaby Leon-Perez, Ying Chao Kao, Mehmet Gurses, Katie Zaman, Maria Azocar, Di Wang, Madeline Pape, Carrie Hough, and the entire Migration Politics editorial college. Many thanks to the reviewers for their insightful comments on the manuscript. I also want to extend my appreciation to the University of Amsterdam for hosting me as a scholar-in-residence during the manuscript's preparation and the support of my MP fellowship cohort, Drs. Christiane Froelhich, Andonea Jon Dickson, Kidjie Saguin, Lea Müller-Funk, and Richa Shivakoti. Last, but never least, I extend my deepest appreciate to my husband Faisal El Anzaoui who always does whatever it takes so that I can pursue my passion and work.

**Funding information. Funding for manuscript writing was provided by the University of Amsterdam and the Migration Politics Fellowship**.

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
