# Peer review of "Moral Lines of Credit: Forging Race Projects, Citizenship, and Nation on Online U.S. Spousal Reunification Forum"

_Migration Politics_

## Round 1 · Author Response

REVISION MEMO

MORAL LINES OF CREDIT: FORGING RACE PROJECTS, CITIZENSHIP, AND NATION ON ONLINE U.S. SPOUSAL REUNIFICATION FORUMS

I thank the reviewers and the editor-in-charge for the thoughtful feedback and commentary on this manuscript. I have seriously considered and implemented your comments and feedback. They significantly improved the paper. Below, I detail how I addressed the suggestions. Please note that there is no Reviewer Two. Reviewer Three's comments were posted twice, which made that individual Reviewer Two and Reviewer Three from my dashboard. I refer to this individual as Reviewer Three throughout the revision memo.

1. Reviewer One requested that I expand on some terms that non-specialist readers may not know. I have cited Bonjour and De Hart using their 2021 piece as asked. On page seven, I discussed the moral line of credit more clearly. Further, I defined "technologies of love" on page six of the manuscript, "bezness" on page seventeen, and "intimate citizenship" on page twenty-six. Reviewer One recommended that I recognize the breadth of literature on family reunification within the introduction. While I found it challenging to work it into the opening, I did acknowledge the vast literature in a detailed footnote. Please see page four for footnote one.

I sincerely appreciate Reviewer One's suggestions on the methods section of the paper, as I see that some of my wording is unclear. Thus, I expanded on my time doing ethnographic observation and why I found Immigration Pathways the ideal choice for my data collection. Further, I reworded 'respondents,' so it does not imply that I spoke with any forum member. I meant the people who responded to the original author of the thread, so this is now corrected. See pages eleven to thirteen for details.

Lastly, I did address the minor issues of citation. I have now put quotation marks around bezness (please see pages sixteen and nineteen). I also made the moral line of credit singular in all instances, except in the title. My rationale for the title is that it refers to every citizen's moral line of credit. In contrast, in the manuscript, I am referring to individual instances where someone is drawing on their moral line of credit. Thank you for keeping me consistent. I also removed several additional spaces and extra punctuation within the paper (see pages one, four, six, seventeen, twenty-five, and twenty-eight. I replaced the & symbol with 'and' (see page seven) and removed the dash on '-undoubtedly skewed' on page twenty-eight. I formalized 'wouldn't' with 'would not' and fixed 'andusing' on page thirty-two. Finally, I correctly cited the missing works in the reference section (i.e., USCIS 2013; Moynihan Report 1965, and Collins 2005, which I meant to be Hill-Collins).

2. Reviewer Three asked for some minor but meaningful revisions. In particular, they asked me to explain and problematize "hegemonic family norms." I expanded on this definition on page eight. I also included that this is a normative understanding of family to emphasize its problematic nature. On page twelve in the Methods section, I briefly explained how the closed coding would reveal how forum members constructed their understandings of family. I chose not to emphasize that these were simply normative iterations of the family because I did look for ways that members pushed back on hegemonic family norms. I felt that saying I looked only for their normative understandings of the family would be equivalent to selecting on the bias. Reviewer Three also requested that the article's title and research question focus specifically on the MENA region and Sub-Saharan Africa. I appreciate this insight and gave it a great deal of thought. However, ultimately, I decided to keep the title and research question the same. I felt that focusing the inquiry on the two specific regions might be limiting. My rationale is that I want to invite future research to examine how citizens' moral lines of credit operate in other regional spaces and/or across different states' immigration systems.

3. Reviewer Four made multiple minor revision requests, which I have satisfied. I referenced the statement "the state is interested in upholding its racial projects" on page three. On page four, I revised the conversation about "irrevocable rights" so that it makes more sense to the reader, and I edited the discussion about "moral gatekeepers" (Wray 2006) so the state does not appear to be a single entity with one, coherent objective. Next, I added a brief description of the citizens' moral currency under the discussion of the political 'economy' on page five. I rephrased for nuance and clarity the discussion on page six regarding citizens' marriage to non-citizens and its interference with state interests. I especially appreciated the request to discuss 'deservingness' in more detail and the literature suggestions. I have done this using some of those readings. Please see page five for this change. Throughout the discussion of the U.S. racial project and its hierarchy on pages five through eleven, I added key historical moments and legislation to contextualize this discussion. I agree with the reviewer that paralleling Arab Americans drawing on social relationships to counteract the 9/11 discourse is too bold a statement. Thus, I revised the language to ease the direct correlation. However, I did not remove this section entirely as I felt that the modified portion is an accurate reflection. Lastly, on page fifteen, I removed the Hill Collins 2005 and Macklin 2022 statement about white women promulgating racial purity through their children and interracial relationships running counter to the state's interests.

I hope that these changes have satisfied the reviewers' requests. Again, thank you for your time and careful attention to this matter. If you have further suggestions or questions about the manuscript, please do not hesitate to contact me via the editor-in-charge.

---

## Round 2 · Author Response

REVISION MEMO

MORAL LINES OF CREDIT: FORGING RACE PROJECTS, CITIZENSHIP, AND NATION ON ONLINE U.S. SPOUSAL REUNIFICATION FORUMS

I thank the editor-in-charge for the thoughtful feedback and commentary on this manuscript. I have seriously considered and implemented your comments and feedback. They significantly improved the paper.

  1. On page 2, I reworded the vague “these requirements” on page two with a clearer sentence that directly connects them to “valid and subsisting.”
  2. On page 4, I have acknowledged the other facets of literature on family reunification. It has been included in the area where I discuss potential contributions and future research. I hope that this is satisfactory.
  3. On page 4, the editor was unclear what I meant by “citizenship was fixed” on page 4. I have revised this sentence now. By fixed I meant that the definition of who could qualify for citizenship was highly limited and selective.
  4. Page 6 had the mistype of ‘r currency’ when it should have read as ‘a currency’. I have remedied this typo
  5. The editor requested that I make the data for this paper public. Unfortunately, I cannot. According to Virginia Commonwealth University’s Internal Review Board requirement for this research project, I am unable to publicly disclose the data due to potential privacy/confidentiality harm risks to the end users (IRB ID# HM20021701). I am willing, however, to make my contact information available to allow for inquiries on individual bases.
  6. On page 11, I expanded on how forum members report their demographic information. Sometimes, they do so in their profile, through their conversations with other members or both. Similarly, at the bottom of that page, I explain that a bystander is someone who signs ups and neither posts nor completes any profile information. Bystanders are “lurking” meaning that you can see by their profile that they log in often, but do not do interact with others. They do not have a real impact on the other members, so I can simply remove this sentence if the editor prefers. I did want to fully disclose what I know of who is on the site, which is why I initially included them.
  7. I have added the month and years that I was ethnographically immersed into the manuscript on page 12. Subsequently, I added how I watched these conversations unfold in real time rather than scrape them and read them later. This is important because each time a thread was replied to, it was “bumped” up to the top of the page…meaning that users would see that thread first. Some threads could linger for months and eventually meander away from the original topic. More important topics would often last a few days maximum and stay on topic. Sometimes, moderators would “lock” a topic, meaning that one could no longer post on it. This happened when someone was being verbally assaulted beyond what was deemed reasonable according to moderators. I spent various times of day and night on the forums, so I knew who was on and when. I could add this to a footnote if you would like, but wasn’t sure how much information the editor would want given the length of the paper.
  8. On page 12, I reworded the paragraph that starts with “I chose Immigration Pathways” for clarity.
  9. On page 12, I changed ‘consist of’ to contain. Moreover, I discuss how my ethnographic immersion helped me formulate the criteria for the purposive sample.
  10. I fixed the block quotations on page 14 so that it would not break off in the middle.
  11. Endnote one can be found in the opening quote after the pseudonym name and “Immigration Pathways”
  12. I have corrected Dr. Fröhlich’s name on page 34.
  13. I have removed endnote ii and have integrated it into the introduction.

I hope that these changes have satisfied your requests. Again, thank you for your time and careful attention to this matter. If you have further suggestions or questions about the manuscript, please do not hesitate to contact me.

---

## Round 2 · List of Changes

REVISION MEMO

MORAL LINES OF CREDIT: FORGING RACE PROJECTS, CITIZENSHIP, AND NATION ON ONLINE U.S. SPOUSAL REUNIFICATION FORUMS

I thank the editor-in-charge for the thoughtful feedback and commentary on this manuscript. I have seriously considered and implemented your comments and feedback. They significantly improved the paper.

1. On page 2, I reworded the vague “these requirements” on page two with a clearer sentence that directly connects them to “valid and subsisting.”
2. On page 4, I have acknowledged the other facets of literature on family reunification. It has been included in the area where I discuss potential contributions and future research. I hope that this is satisfactory.
3. On page 4, the editor was unclear what I meant by “citizenship was fixed” on page 4. I have revised this sentence now. By fixed I meant that the definition of who could qualify for citizenship was highly limited and selective.
4. Page 6 had the mistype of ‘r currency’ when it should have read as ‘a currency’. I have remedied this typo
5. The editor requested that I make the data for this paper public. Unfortunately, I cannot. According to Virginia Commonwealth University’s Internal Review Board requirement for this research project, I am unable to publicly disclose the data due to potential privacy/confidentiality harm risks to the end users (IRB ID# HM20021701). I am willing, however, to make my contact information available to allow for inquiries on individual bases.
6. On page 11, I expanded on how forum members report their demographic information. Sometimes, they do so in their profile, through their conversations with other members or both. Similarly, at the bottom of that page, I explain that a bystander is someone who signs ups and neither posts nor completes any profile information. Bystanders are “lurking” meaning that you can see by their profile that they log in often, but do not do interact with others. They do not have a real impact on the other members, so I can simply remove this sentence if the editor prefers. I did want to fully disclose what I know of who is on the site, which is why I initially included them.
7. I have added the month and years that I was ethnographically immersed into the manuscript on page 12. Subsequently, I added how I watched these conversations unfold in real time rather than scrape them and read them later. This is important because each time a thread was replied to, it was “bumped” up to the top of the page…meaning that users would see that thread first. Some threads could linger for months and eventually meander away from the original topic. More important topics would often last a few days maximum and stay on topic. Sometimes, moderators would “lock” a topic, meaning that one could no longer post on it. This happened when someone was being verbally assaulted beyond what was deemed reasonable according to moderators. I spent various times of day and night on the forums, so I knew who was on and when. I could add this to a footnote if you would like, but wasn’t sure how much information the editor would want given the length of the paper.
8. On page 12, I reworded the paragraph that starts with “I chose Immigration Pathways” for clarity.
9. On page 12, I changed ‘consist of’ to contain. Moreover, I discuss how my ethnographic immersion helped me formulate the criteria for the purposive sample.
10. I fixed the block quotations on page 14 so that it would not break off in the middle.
11. Endnote one can be found in the opening quote after the pseudonym name and “Immigration Pathways”
12. I have corrected Dr. Fröhlich’s name on page 34.
13. I have removed endnote ii and have integrated it into the introduction.

I hope that these changes have satisfied your requests. Again, thank you for your time and careful attention to this matter. If you have further suggestions or questions about the manuscript, please do not hesitate to contact me.

---

## Editorial Decision

unknown